# Musculoskeletal Ultrasound in Monitoring Clinical Response to Treatment in Acute Symptomatic Psoriatic Dactylitis: Results from a Multicentre Prospective Observational Study

**DOI:** 10.3390/jcm9103127

**Published:** 2020-09-27

**Authors:** Nicolò Girolimetto, Pierluigi Macchioni, Niccolò Possemato, Ilaria Tinazzi, Vittoria Bascherini, Giorgia Citriniti, Rebecca McConnell, Antonio Marchetta, Rosario Peluso, Vincenzo Sabbatino, Carlo Salvarani, Raffaele Scarpa, Luisa Costa, Francesco Caso

**Affiliations:** 1Department of Clinical and Experimental Medicine, Rheumatology Research Unit, University Federico II, 80138 Naples, Italy; vittoria.bascherini@hotmail.it (V.B.); rosario.peluso2@unina.it (R.P.); sabbatino.vincenzo2@gmail.com (V.S.); raffaele.scarpa@unina.it (R.S.); lv.costa@libero.it (L.C.); francescocaso1@yahoo.it (F.C.); 2Department of Rheumatology, Azienda USL-IRCCS di Reggio Emilia, 42122 Reggio Emilia, Italy; pierluigi.macchioni@ausl.re.it (P.M.); niccolo.possemato@ausl.re.it (N.P.); giorgia.citriniti@gmail.com (G.C.); carlo.salvarani@ausl.re.it (C.S.); 3Unit of Rheumatology, IRCCS Sacro Cuore Don Calabria Hospital, 37024 Negrar, Italy; ilariatinazzi@yahoo.it (I.T.); antonio.marchetta@sacrocuore.it (A.M.); 4Neurosciences Center, Bronson Health Group, Kalamazoo, MI 49007, USA; 2rebeccamcconnell@gmail.com; 5Rheumatology Unit, University of Modena and Reggio Emilia, 41121 Modena, Italy

**Keywords:** dactylitis, psoriatic arthritis, ultrasound, flexor tenosynovitis, soft tissue odema

## Abstract

This observational and prospective study evaluated the clinical correlations of sonographic lesions in consecutive psoriatic arthritis (PsA) dactylitis cases. Eighty-three dactylitic digits were evaluated clinically and sonographically before treatment and at one-month (T1) and three-month (T3) follow-up. Clinical evaluation included the Leeds Dactylitis Index-basic (LDI-b) score and the visual analogue scales for pain (VAS-p) and functional impairment (VAS-FI). High-frequency ultrasound with grey scale (GS) and power Doppler (PD) assessed flexor tenosynovitis (FT), soft tissue oedema (STO), extensor tendon paratenonitis, and joint synovitis. There was a statistically significant correlation between the clinical parameters (VAS-p, VAS-FI, and LDI-b) and FT and STO at T1 and T3. We found statistically significant improvement in FT and STO for the cases with clinically meaningful treatment responses (*p* < 0.001). After a multiple conditional logistic regression analysis, the only variables that correlated with a T1 clinical response were the resolutions of PD FT (OR 15.66) and PD STO (OR 6.23), while the resolution of PD FT (OR 27.77) and of GS STO (OR 7.29) correlated with a T3 clinical response. The clinical improvements of active dactylitis are linked to the regression of sonographic evidence of extracapsular inflammation (particularly FT and STO).

## 1. Introduction

Dactylitis is a pathognomonic manifestation of psoriatic arthritis (PsA) and a diagnostic criterion in the ClASsification for Psoriatic ARthritis (CASPAR) [1,2]. Its presence, characterised by diffuse swelling of a whole digit, can assist in disease diagnosis and serve as a clinical marker of disease severity [3,4]. Dactylitis occurs in 16 to 49% of PsA patients, and it may be the first and only pathological manifestation for many years [5]. The measurement of dactylitis severity varies from a simple count of the affected fingers to the Leeds Dactylitis Index (LDI) and LDI-basic (LDI-b), which evaluate finger circumference and tenderness [6,7]. The Group for Research and Assessment of Psoriasis and PsA (GRAPPA) [8] and the European League Against Rheumatism (EULAR) [9] recommend non-steroidal anti-inflammatory drugs (NSAIDs) and local corticosteroid injections for symptomatic management.

Dactylitis has variable associations to inflammatory lesions, including flexor tenosynovitis (FT), soft tissue oedema (STO), and joint synovitis [10]. However, these pathophysiological features of dactylitis can only be appreciated with advanced imaging with musculoskeletal-ultrasound (Msk-US) and magnetic resonance imaging (MRI) studies [11,12,13,14]. Compared to rheumatoid arthritis (RA), psoriasis, and healthy controls, PsA fingers demonstrate a high prevalence of US-determined periarticular soft tissue alterations, particularly at flexor tendon pulleys [15,16,17]. Recent studies reported significant extracapsular inflammation (FT and STO) in early dactylitis and a higher prevalence of joint synovitis at the proximal interphalangeal (PIP) level in chronic cases [18,19,20,21]. Additional cross-sectional studies linked specific US lesions (particularly FT and STO) with local, patient symptoms [18,19,20,21]. Moreover, a recent study showed that high values of LDI-b (indicating more disease) are associated with FT and STO on US [21]. The results of these cross-sectional studies suggest that FT and STO could play a role in triggering and sustaining the local symptoms of dactylitis.

Until this study, no longitudinal investigations have analysed the associations between the clinical dactylitic activity and the US features of PsA dactylitis before and after treatment. The aim of the present study is to explore the associations between specific US findings and clinical parameters of disease activity in a consecutive series of acute dactylitic episodes of PsA patients over 3 months.

## 2. Experimental Section

Three Italian rheumatology centres (i.e., Rheumatology Unit in Naples, Reggio Emilia, and Negrar) enrolled consecutive PsA patients with symptomatic hand dactylitis. The study was approved by the local ethical committees of the participating centres, and a written, informed consent was obtained from all participants.

The inclusion criteria were the following: (1) adult patients (18 years of age or older) affected by PsA satisfying CASPAR criteria; (2) symptomatic hand dactylitis; (3) otherwise well-controlled PsA. The exclusion criteria were the following: (1) previous episodes of dactylitis in the involved finger; (2) previous corticosteroid injection in the involved finger; (3) current use of biological disease-modifying antirheumatic drugs (bDMARDs). If an enrolled patient developed symptoms from an additional finger, the newly symptomatic digit was assessed at baseline and represented a new case of dactylitis in the study.

The presence of active dactylitis was diagnosed and evaluated by one rheumatologist from each centre and confirmed using a dactylometer and the LDI-b [6]. The LDI-b measurement starts with the ratio between the circumference of the involved finger and the contralateral, non-affected one; a difference of more than 10% is sufficient to confirm the dactylitis. The circumference ratio is multiplied by a tenderness score (0 for non-tender, 1 for tender) to obtain an LDI-b score [7]. For each patient, the Disease Activity Index for Psoriatic Arthritis (DAPSA) score was calculated at baseline [22,23]. If the disease was well-controlled, apart from active dactylitis, patients qualified for study enrolment and continued the same baseline treatment during the entire follow-up period. After baseline examination (T0), patients were treated either with local steroid injections or with oral non-steroidal anti-inflammatory drugs (NSAIDs) based on patient preference, in accordance with routine clinical practice and local guidelines [24]. The injection was performed with a 25-gauge needle introducing 20 mg of methylprednisolone acetate into the flexor tendon sheath.

Dactylitis occurring at different times during the enrolment period, even if occurring in the same patient, represented a new case of dactylitis included in the study. A new case was included only if the previous episode of dactylitis was resolved and had completed treatment

### 2.1. Clinical Assesment

The patients rated their local pain severity and functional impairment of the symptomatic digit using a Visual Analogue Scale (VAS). Pain (VAS-p) scores ranged from 0 (no pain) to 10 (worst pain imaginable), [25] and functional impairment (VAS-FI) ranged from 0 (no functional impairment) to 10 (complete functional impairment) [26]. Outcomes were evaluated at baseline and at the first and third month post-treatment (T1 and T3, respectively) by an independent clinical assessor (one from each centre). Clinical evaluation included physical examination of the affected finger, measurement of finger circumference using the dactylometer, calculation of the LDI-b, and the patient’s VAS-p and VAS-FI scores. Local and systemic adverse events were also recorded at every clinical examination: T0, T1, and T3.

We considered the primary objective to be achieved if the affected finger had a clinically meaningful treatment response (MTR). We defined MTR as a reduction of at least 5 points in VAS-p and VAS-FI or if both VAS-p and VAS-FI scores were < 2.

### 2.2. Ultrasound Assesment

Sonographic evaluations of the dactylitic finger were performed by three rheumatologists (NG, PM, IT), experts in Msk-US, who were blinded to the clinical evaluations. All Msk-US scans were performed in the same setting using the same equipment model (MyLab70XVG—Esaote S.p.A., Genoa, Italy machine, with a 6–18 MHz linear transducer). The ultrasound grey scale (GS) imaging parameters were optimised for maximal image resolution. Power Doppler (PD) was standardised with a 500 Hz pulse repetition frequency, 3 wall filter, 4 persistence, and 45–55% colour gain. Msk-US examinations used the appropriate quantity of gel to avoid probe contact with the skin. The dactylitic digit and the contralateral, non-affected finger were scanned from the dorsal and volar views in longitudinal and transverse planes per currently accepted international guidelines [27]. The non-affected finger was scanned to allow for STO grading in the dactylitic finger.

The following dactylitis-related sonographic lesions were investigated by GS and PD: FT, STO, peritendon extensor inflammation (PTI) at the metacarpophalangeal (MCP) and proximal interphalangeal (PIP) joint levels, and joint synovitis at the MCP, PIP and distal interphalangeal (DIP) joints. FT was assessed in GS and PD modes using the four-point, semi-quantitative scoring scale proposed by the Outcome Measures in Rheumatology (OMERACT) US group [28]. As described in a recent study, STO was defined as “abnormal hypoechoic/anechoic areas, diffused or localized within the subcutaneous tissue between the epidermis and the tendon-related anatomic structures, with local thickening, with or without local abnormal Doppler signal, visualized in two perpendicular planes, and not present on the contralateral side.” [29,30]. STO, therefore, was a comparative evaluation between the affected and un-affected contralateral finger with a semi-quantitative (0–3) score in GS and PD modes: grade 0, no finger oedema; grade 1, mild; grade 2, moderate; grade 3, severe [29,30]. PTI, identified by a hypoechoic area surrounding a tendon without a synovial sheath (with or without peri-tendinous PD signal), was evaluated dichotomously (0–1) in GS and PD mode at the MCP and PIP joints level, as previously reported [29]. We used OMERACT’s definition of synovitis as any grade of PD signal and hypoechoic synovial hypertrophy, regardless of effusion [31,32]. Joint synovitis was scored using the EULAR–OMERACT combined score for GS and PD (0–3) [32,33]. All symptomatic digits underwent US examination at T0, T1, and T3.

### 2.3. Statistical Analysis

The statistical analyses were performed using IBM SPSS Statistics for Windows, Version 23 (IBM Corp., Armonk, NY, USA). All quantitative variables were expressed in terms of mean ± SD or, in cases of strong violation of normality, median and range. Qualitative variables were expressed as percentages. When appropriate, continuous variables were compared using the t-test or non-parametric tests, and non-continuous variables were compared using the Chi-square test. Correlations between variables were calculated with Spearman’s rho. Statistical tests were performed at a significance level of α = 0.05. All demographic, clinical, and laboratory variables were entered as possible explanatory variables in a conditional logistic regression analysis with each response at T1 and T3 examination as the dependent variable. The most significant independent variables were identified using a *p*-value greater than 0.10 as the removal criterion using a backward selection procedure.

Adequate intra- and inter-rater reliability were assumed based on significant agreement reported in previous studies [18,19,20,21,29].

Sensitivity to change of individual US lesions was estimated using the standardized response mean (SRM). The SRM is the mean change score divided by the standard deviation of the change and was stratified as trivial < 0.20, small 0.20–0.40, moderate 0.50–0.79 and good > 0.80 [34]. The association between the changes in US lesions and clinical variables was assessed by a mixed model for repeated measurement.

## 3. Results

### 3.1. Patients’ Characteristics and Clinical Findings

Eighty-three symptomatic, dacylitic hand digits from 56 PsA patients entered the study (23 female and 33 male; mean age 49.2 ± 13.8 years; PsA duration 49.55 ± 44.4 months). Fifteen patients developed more than one episode of dactylitis; no patient presented with two or more symptomatic fingers simultaneously nor had multiple episodes involving the same finger. All enrolled patients completed T0, T1, and T3 evaluations for every involved digit. All patients had well-controlled disease apart from active dactylitis (DAPSA < 12) and continued the same baseline treatment (oral steroids, conventional synthetic DMARDs—csDMARDs—or no treatment) throughout the study. Patients did not show axial involvement. Since dactylitis was the only active manifestation of PsA, patients were only treated with a local steroid injection or oral NSAIDs, according to local and international recommendations [8,9,24]. Forty-two fingers were injected with local steroid, while 41 were managed with NSAIDs (ibuprofen or etoricoxib).

Table 1 summarises patient and dactylitis characteristics at baseline.

According to our definition of MTR, 54 out of 83 cases of dactylitis were treatment responders at T3. There were no relapses of dactylitis in responders at 3 months.

### 3.2. Ultrasound Findings

During the follow-up period, we observed a significant reduction in FT and STO at T1 and T3 examinations (see Table 2). PTI and joint synovitis did not change significantly. We observed significantly less FT and STO (both in GS and in PD) at T1 and T3 in the MTR group (responders) compared to the non-MTR group (Table 3). The prevalence of the other US parameters (PTI and joint synovitis) did not reach statistical significance.

### 3.3. Correlation between Ultrasound and Clinical Parameters

We observed a statistically significant correlation (Spearman) between the clinical parameters (VAS-p, VAS-FI and LDI-b) and the US signs of flexor tendon and soft tissue involvement at T1 and T3 examinations (see Table 4). We did not observe a clinical correlation with PTI or synovitis at the MCP, PIP, or DIP.

Msk-US assessment demonstrated a significant reduction in some US lesions after treatment at T1 examination, confirmed at T3 examination. The SRM between T0 and T1 and T0 and T3 evaluations were good for GS FT (0.91 and 1.32) and PD FT (0.92 and 0.99), moderate for GS STO (0.72 and 0.67) and PD STO (0.70 and 0.71), and trivial for the remaining lesions (0–0.17).

Changes of FT and STO (both in GS and PD) were significantly associated with changes in VAS-pain, VAS-FI and LDI-b, using a mixed model for repeated measurement (*p* < 0.001).

The univariate logistic regression analysis did not show associations between the clinical response at T1 and T3 with patient demographics or baseline clinical or US findings. The only variables significantly associated with clinical response at T1 were GS FT (OR 3.64 (95%CI: 1.05; 12.65) *p* < 0.001), PD FT (OR 14.67 (95%CI: 3.59–59.88) *p* < 0.001), GS STO (OR 8.65 (95%CI: 2.23–33.59) *p* = 0.002), and PD STO (OR 6.0 (95%CI: 1.52–23.64) *p* = 0.009). The same variables were associated with the clinical response at T3: GS FT (OR 13.3 (95%CI: 2.75; 63.92) *p* < 0.001), PD FT (OR 48.0 (95%CI: 8.75–263.1) *p* < 0.001), GS STO (OR 15.97 (95%CI: 3.33–76.53) *p* = 0.001), and PD STO (OR 6.63 (95%CI: 1.71–25.77) *p* = 0.017).

When these variables were entered in a multiple conditional logistic regression analysis, only the resolution of PD FT (1.7 OR (1.1–3.1) *p* = 0.045) and GS STO (2.5 OR (1.3–4.8) *p* = 0.006) significantly improved at T1, whereas the resolution of PD FT (OR 2.3 (1.2–4.4) *p* = 0.010) and of GS STO (OR 2.8 (1.4–5.2) *p* = 0.002) was associated with T3. Neither GS nor PD changes of US-determined synovitis (MCP, PIP and DIP) or PTI correlated with clinical responses.

An example of the US assessment at baseline and after treatment in the MTR group is described in Figure 1.

## 4. Discussion

This study is a long-awaited, longitudinal assessment of PsA dactylitis with Msk-US and clinical indices. Among the most pathognomonic US elementary lesions evaluated, FT and STO showed good sensitivity to change and reflected the clinical response at three months of patients with PsA dactylitis treated with a steroid injection or NSAIDs.

The sonographic improvement appeared to be more significant if we compared the individual US lesions between the MTR group (responders) and the non-MTR group. At the three-month follow-up, resolution of FT and STO (both in GS and PD) correlated with an improvement in VAS-p, VAS-FI and LDI-b. On the other hand, the resolution of synovitis and PTI did not correlate with clinical improvement. These observations are consistent with the results of our previous, cross-sectional studies.

The link between sonographic features of dactylitis in PsA and local symptoms were evaluated for the first time in a recent study where local pain and tenderness were positively associated with GS FT (grade ≥ 2), PD FT, GS STO, and PD STO (*p* < 0.001 for all comparisons) [18]. Moreover, there was a negative association with joint synovitis (*p* < 0.001, both in GS and PD). The same positive and negative associations with sonographic findings were found in a cohort of 100 cases of dactylitis regardless of whether the symptom duration was greater or less than 20 weeks (*p* < 0.001 for all comparisons). The data from this cohort demonstrated that extracapsular inflammation predominates in the early phases of dactylitic episodes. Cases with a shorter dactylitis duration (<20 weeks) had a significantly higher prevalence of GS FT (grade ≥ 2), PD FT, GS STO, and PD STO (*p* = 0.001, *p* < 0.001, *p* < 0.05, and *p* = 0.001, respectively) [19]. However, synovitis at the PIP joint (identified in GS and PD) was more frequent in patients with longer dactylitis duration (*p* < 0.001).

In another publication, we compared the clinical and sonographic features of 80 symptomatic with 36 asymptomatic cases of PsA dactylitis. Symptomatic fingers had a significantly shorter dactylitis duration (more acute presentation) compared to asymptomatic fingers (*p* < 0.001) [20]. Values indicative of clinical presentation—LDI, patient VAS-p, and VAS-FI—were significantly higher in fingers with symptomatic dactylitis (*p* < 0.001, *p* < 0.001, and *p* = 0.010, respectively). Symptomatic dactylitis had a higher prevalence of FT grade > 2 and STO, both in GS and PD modes (*p* < 0.001). Asymptomatic dactylitis showed a greater prevalence of joint synovitis in GS and PD modes (*p* < 0.001). Due to the clinical utility of the LDI-b, we recently explored the link between specific sonographic findings and the LDI-b score of 91 cases of hand dactylitis in 63 PsA patients. [21] Digits with an LDI-b score above the median had significantly higher prevalence of GS FT, PD FT and STO (*p* = 0.015, *p* = 0.001, and *p* = 0.004, respectively). From these cross-sectional studies, we deduce that pain and digital tenderness are linked to dactylitis duration, and earlier lesions are associated with extra-articular inflammatory changes, particularly FT and STO. These findings suggest an extra-articular basis for symptoms in PsA dactylitis.

Accessory pulley involvement is another extra-articular manifestation linked to PsA compared to RA, psoriasis, and healthy controls. This trend was especially notable in subjects with a history of dactylitis [16]. These data were confirmed in a sonographic study of 58 PsA patients. This study showed increased PD signal of the intra-pulley and thickening of pulleys A1, A2 and A4 in the dactylitic digits [17]. Additionally, almost all of the dactylitic digits showed peri-tendinous oedema, and 82.6% showed FT with PD signals in 65.2% of flexor tendons. These data substantiate the pivotal role of tenosynovitis and extra-articular disease in symptomatic dactylitis.

Recently, a score (DACTylitis glObal Sonographic—DACTOS) for grading the characteristic US lesions in PsA hand dactylitis was developed [30]. This composite score includes FT, STO, PTI and joint synovitis, both in GS and PD modes. It would be interesting to apply this score in a longitudinal study to evaluate the response to treatment and explore the possible correlation with clinical parameters of disease activity.

This study had a number of limitations. First, we included more than one dactylitic finger from fifteen of the enrolled patients. Each episode of dactylitis in these patients occurred at different times and involved different fingers. We chose to include these cases because we could identify a unique set of clinical and sonographic characteristics in each episode of dactylitis, even when occurring in the same patient. Another limitation is the relatively short follow-up period, which should be prolonged to understand the progression of this chronic disease and discover favourable sonographic treatment indicators. At the moment, a commonly shared definition of meaningful treatment response (MTR) for dactylitis is not available, and as a result, we decided to define MTR based on our clinical practice. For a patient’s improvement to be clinically meaningful, we hoped to see a simultaneous reduction of at least 5 points in VAS-pain and in VAS-FI, or both VAS-pain and VAS-FI scores of less than 2. Furthermore, this study does not have a placebo arm. We cannot exclude that local infiltration may have had a placebo effect on patient-reported symptoms. However, the number of patients treated with infiltration was approximately equal to those treated with NSAIDs. To better assess the objective response to treatment, we utilized US evaluation, which is not affected by the patient’s opinion. Finally, our current results need additional validation with data from a more robust case series.

## 5. Conclusions

To our knowledge, this is the first longitudinal study that persuasively demonstrates the link between clinical response in dactylitis and sonographic improvement in extracapsular lesions (particularly FT and STO). These findings confirm the central role of extra-articular structures as the pivotal epicentres of local inflammation in active dactylitis.

## 6. Patents

No patents resulting from the work reported in this manuscript.

## Figures and Tables

**Figure 1 jcm-09-03127-f001:**
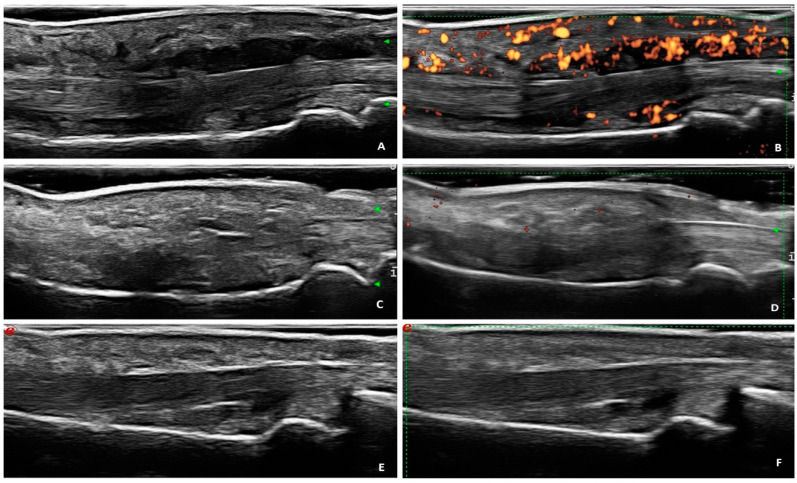
(**A**,**B**) Volar scan of a dactylitic digit showing flexor tenosynovitis (grade 3 for GS and PD) and soft tissue oedema (grade 2 for GS and PD) at baseline. (**C**,**D**) Longitudinal views of the contralateral, unaffected digit. (**E**,**F**) Volar scan of dactylitic digit after three months showing the absence of flexor tenosynovitis and soft tissue oedema (both in GS and PD. Dorsal scans are not shown because joint synovitis and peritendon extensor inflammation were not present. Abbreviations: GS: grey scale; PD: power Doppler.

**Table 1 jcm-09-03127-t001:** Baseline demographics and clinical characteristics of 56 patients and 83 dactylitic fingers.

**Patients (56)**	
Female *n* (%)	23 (41%)
Mean age, years (mean ± SD)	49.2 + 13.8
**Dactylitic fingers (83)**	
Dactylitis duration, weeks (mean ± SD)	23.3±22.3
PsA disease duration at dactylitic episode, months (mean ± SD)	49.5 ± 44.4
Psoriasis duration at dactylitic episode, months (mean ± SD)	69.3 ± 52.8
ESR, mm/hr (mean ± SD)	18.9 ± 13.3
CRP, mg/dl (mean ± SD)	1.4 ± 1.8
TJC 68 joints (mean ± SD)	7.5 ± 4.2
SJC 66 joints (mean ± SD)	2.8 ± 2.9
Nail involvement (*n* (%))	59 (71%)
Presence of at least one enthesitis (*n* (%))	43 (52%)
Enthesitis count (mean ± SD)	1.1 ± 1.3
**Therapy at baseline**	
No therapy (*n* (%))	27 (32%)
csDMARDs (*n* (%))	53 (64%)
Oral steroids (*n* (%))	3 (4%)

Abbreviations: CRP: C-reactive protein; csDMARDs: conventional synthetic disease-modifying antirheumatic drugs; ESR: erythrocyte sedimentation rate; PsA: psoriatic arthritis; SJC: swollen joint count; TJC: tender joint count.

**Table 2 jcm-09-03127-t002:** Prevalence of ultrasound abnormalities observed in 83 dactylitic fingers from 49 psoriatic arthritis patients evaluated at baseline, 1 month, and 3 months.

Variable	Grade 0	Grade 1	Grade 2	Grade 3	*p*-Value
**GS flexor tenosynovitis**					
T0	6 (7.1%)	26 (31.0%)	45 (53.6%)	7 (8.3%)	
T1	24 (28.6%)	41 (48.8%)	16 (19.0%)	3 (3.6%)	T0 vs. T1; *p* < 0.001
T3	41 (48.8%)	29 (34.5%)	13 (15.5%)	1 (1.2%)	T0 vs. T3; *p* < 0.001
**PD flexor tenosynovitis**					
T0	16 (19.0%)	9 (10.7%)	51 (60.7%)	8 (9.5%)	
T1	45 (53.6%)	19 (22.6%)	16 (19.0%)	4 (4.8%)	T0 vs. T1; *p* < 0.001
T3	56 (66.7%)	10 (11.9%)	17 (20.2%)	1 (1.2%)	T0 vs. T3; *p* < 0.001
**GS soft tissue oedema**					
T0	4 (4.8%)	32 (38.1%)	38 (45.2%)	10 (11.9%)	
T1	25 (29.8%)	39 (46.4%)	17 (20.2%)	3 (3.6%)	T0 vs. T1; *p* < 0.001
T3	35 (41.7%)	29 (34.5%)	16 (19.0%)	4 (4.8%)	T0 vs. T3; *p* < 0.001
**PD soft tissue oedema**					
T0	8 (9.5%)	25 (29.8%)	42 (50.0%)	9 (10.7%)	
T1	27 (32.1%)	29 (34.5%)	26 (31.0%)	2 (2.4%)	T0 vs. T1; *p* < 0.001
T3	33 (39.3%)	27 (32.1%)	22 (26.2%)	2 (2.4%)	T0 vs. T3; *p* < 0.001
**MCP GS** **peritendon extensor inflammation**					
T0	69 (94.5%)	4 (5.5%)			
T1	69 (94.5%)	4 (5.5%)			T0 vs. T1; *p* = ns
T3	70 (95.8%)	3 (4.2%)			T0 vs. T3; *p* = ns
**MCP PD** **peritendon extensor inflammation**					
T0	81 (96.4%)	2 (2.4%)	1 (1.2%)		
T1	81 (96.4%)	3 (3.6%)			T0 vs. T1; *p* = ns
T3	82 (97.2%)	2 (2.4%)			T0 vs. T3; *p* = ns
**PIP GS** **peritendon extensor inflammation**					
T0	71 (97.3%)	2 (2.7%)			
T1	71 (97.2%)	2 (2.7%)			T0 vs. T1; *p* = ns
T3	72 (98.6%)	1 (1.4%)			T0 vs. T3; *p* = ns
**PIP PD** **peritendon extensor inflammation**					
T0	71 (97.3%)	2 (2.7%)			
T1	72 (98.6%)	1 (1.4%)			T0 vs. T1; *p* = ns
T3	72 (98.6%)	1 (1.4%)			T0 vs. T3; *p* = ns
**MCP synovitis (combined score)**					
T0	70 (89.7%)	4 (5.1%)	1 (1.3%)	3 (3.8%)	
T1	72 (92.3%)	2 (2.6%)	1 (1.3%)	3 (3.8%)	T0 vs. T1; *p* = ns
T3	70 (89.7%)	5 (6.4%)	1 (1.3%)	2 (2.6%)	T0 vs. T3; *p* = ns
**PIP synovitis (combined score)**					
T0	59 (75.6%)	3 (3.8%)	6 (7.7%)	10 (12.8%)	
T1	59 (75.6%)	5 (6.4%)	3 (3.8%)	11 (14.1%)	T0 vs. T1; *p* = ns
T3	60 (76.9%)	1 (1.3%)	9 (11.5%)	8 (10.3%)	T0 vs. T3; *p* = ns
**DIP synovitis (combined score)**					
T0	71 (91.0%)	1 (1.3%)	6 (7.7%)	0	
T1	71 (91.0%)	0	7 (9.0%)	0	T0 vs. T3; *p* = ns
T3	73 (93.6%)	0	5 (6.4%)	0	T0 vs. T1; *p* = ns

Abbreviations: DIP: distal interphalangeal; GS: grey-scale; MCP: metacarpophalangeal; MTR: meaningful treatment response; PD: power Doppler; PIP: proximal interphalangeal. T1: 1 month; T3: 3 months.

**Table 3 jcm-09-03127-t003:** Prevalence of ultrasound abnormalities at 1 month and 3 months according to the patients’ meaningful treatment response (MTR). Peritendon extensor inflammation is not reported due to low case prevalence (*p* not significant).

Variable	Grade 0	Grade 1	Grade 2	Grade 3
**GS flexor tenosynovitis**				
MTR at T1 (46 fingers)	15 (32.6%)	26 (56.5%)	3 (6.5%)	2 (4.3%)
Non MTR at T1 (37 fingers)	9 (24.3%)	14 (37.8%)	13 (35.1%)	1 (2.7%)
*p* = 0.013				
MTR at T3 (54 fingers)	31 (57.4%)	20 (37.0%)	2 (3.7%)	1 (1.9%)
Non MTR at T3 (29 fingers)	10 (34.5%)	8 (27.6%)	11 (37.9%)	0
*p* = 0.001				
**PD flexor tenosynovitis**				
MTR at T1 (46 fingers)	31 (67.4%)	9 (19.6%)	5 (10.9%)	1 (2.2%)
Non MTR at T1 (37 fingers)	14 (37.8%)	9 (23.3%)	11 (29.7%)	3 (8.1%)
*p* = 0.032				
MTR at T3 (54 fingers)	45 (83.5%)	4 (7.4%)	4 (7.4%)	1 (1.9%)
Non MTR at T3 (29 fingers)	11 (37.9%)	5 (17.2%)	13 (44.8%)	0
*p* < 0.001				
**GS soft tissue oedema**				
MTR at T1 (46 fingers)	20 (43.5%)	20 (43.5%)	6 (13.6%)	0
Non MTR at T1 (37 fingers)	5 (13.5%)	18 (48.6%)	11 (29.7%)	3 (8.1%)
*p* = 0.005				
MTR at T3 (54 fingers)	30 (55.6%)	19 (35.2%)	4 (7.4%)	1 (1.9%)
Non MTR at T3 (29 fingers)	5 (17.2%)	9 (31.0%)	12 (41.4%)	3 (10.3%)
*p* < 0.001				
**PD soft tissue oedema**				
MTR at T1 (46 fingers)	18 (39.1%)	15 (32.6%)	13 (28.3%)	0
Non MTR at T1 (37 fingers)	9 (24.3%)	13 (35.1%)	13 (35.1%)	2 (5.4%)
*p* = 0.239				
MTR at T3 (54 fingers)	27 (50.0%)	15 (27.8%)	12 (22.2%)	0
Non MTR at T3 (29 fingers)	4 (20.7%)6	11 (37.9%)	10 (34.5%)	2 (6.9%)
*p* = 0.023				
**MCP synovitis (combined score)**				
MTR at T1 (46 fingers)	44 (95.7%)	2 (4.3%)	0	0
Non MTR at T1 (37 fingers)	32 (86.5%)	1 (2.7%)	1 (2.7%)	3 (8.1%)
*p* = 0.150				
MTR at T3 (54 fingers)	54 (100%)	0	0	0
Non MTR at T3 (29 fingers)	21 (72.4%)	5 (17.2%)	1 (3.4%)	2 (6.9%)
*p* = 0.001				
**PIP synovitis (combined score)**				
MTR at T1 (46 fingers)	37 (80.4%)	5 (10.9%)	2 (4.3%)	2 (4.3%)
Non MTR at T1 (37 fingers)	26 (70.3%)	1 (2.7%)	1 (2.7%)	9 (24.3%)
*p* = 0.037				
MTR at T3 (54 fingers)	43 (79.6%)	1 (1.9%)	5 (9.3%)	5 (9.3%)
Non MTR at T3 (29 fingers)	21 (72.4%)	1 (3.4%)	4 (13.8%)	3 (10.3%)
*p* = 0.871				
**DIP synovitis (combined score)**				
MTR at T1 (46 fingers)	42 (93.5%)	0	3 (6.5%)	0
Non MTR at T1 (37 fingers)	33 (89.2%)	0	4 (10.8%)	0
*p* = 0.485				
MTR at T3 (54 fingers)	51 (94.4%)	0	3 (5.6%)	0
Non MTR at T3 (29 fingers)	27 (93.1%)	0	2 (6.9%)	0
*p* = 0.807				

Abbreviations: DIP: distal interphalangeal; GS: grey scale; MCP: metacarpophalangeal; MTR: meaningful treatment response (>5 point improvement in VAS-p and VAS-FI or VAS-p and VAS-FI <2); PD: power Doppler; PIP: proximal interphalangeal.T1: 1 month; T3: 3 months.

**Table 4 jcm-09-03127-t004:** Correlation between sonographic lesions and clinical parameters at 1 month and 3 months vs. baseline values.

	VAS-p Delta T0 vs. T1	VAS-FI Delta T0 vs. T1	LDI-b Delta T0 vs. T1
**GS FT Delta T0 vs. T1**	0.537 (<0.001)	0.570 (<0.001)	0.467(<0.001)
**PD FT Delta T0 vs. T1**	0.594 (<0.001)	0.579 (<0.001)	0.545 (<0.001)
**GS STO Delta T0 vs. T1**	0.420 (<0.001)	0.500 (<0.001)	0.386 (<0.001)
**PD STO Delta T0 vs. T1**	0.400 (<0.001)	0.410 (<0.001)	0.293 (0.009)
	**VAS-p Delta T0 vs. T3**	**VAS-FI Delta T0 vs. T3**	**LDI-b Delta T0 vs. T3**
**GS FT Delta T0 vs. T3**	0.590 (<0.001)	0.618 (<0.001)	0.548 (<0.001)
**PD FT Delta T0 vs. T3**	0.525 (<0.001)	0.507 (<0.001)	0.482 (<0.001)
**GS STO Delta T0 vs. T3**	0.526 (<0.001)	0.514 (<0.001)	0.462 (<0.001)
**PD STO Delta T0 vs. T3**	0.396 (0.001)	0.383 (0.001)	0.258 (0.023)

Abbreviations: FT: flexor tenosynovitis; GS: grey scale; LDI-b: Leed’s dactylitic index basic; PD: power Doppler; STO: soft tissue oedema; T0: baseline; T1: 1 month; T3: 3 months; VAS-p: visual analogue scale-pain; VAS-FI: Visual analogue scale-functional impairment.

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
