# Peer review of "Musculoskeletal Ultrasound in Monitoring Clinical Response to Treatment in Acute Symptomatic Psoriatic Dactylitis: Results from a Multicentre Prospective Observational Study"

_jcm, 2020, doi:10.3390/jcm9103127_

Round 1

Reviewer 1 Report

Thank you for the amendments and updates on the revised manuscript, I am satisfied that all the comments have been responded to satisfactorily.

Reviewer 2 Report

My comments were addressed

This manuscript is a resubmission of an earlier submission. The following is a list of the peer review reports and author responses from that submission.

Round 1

Reviewer 1 Report

This study explores the longitudinal change in dactylitis in response to local therapy, and compared clinical and sonographic changes. The study is well written, with original findings. I have a few comments that will hopefully improve the manuscript with additional clarification.

Introduction, pg 2: Suggest more recent key dactylitis references for the 2nd paragraph

  • Pathophysiology of dactylitis: McGonagle, et al. Nat Rev Rheumatol 2019 Feb;15(2):113-122
  • MRI of enthesitis in dactylitis: Tan, et al. Ann Rheum Dis 2015 Jan;74(1):185-9

Methods, ultrasound: The explanation for scanning the contralateral non-affected digit should come immediately after mentioning it. You could say something like: '.....accepted international guidelines. The non-affected figner was scanned to allow for STO grading in the dactylitis finger - see below'.

Results 3.1, pg 4: Please can you clarify how many digits were injected with steroids and the dosages used, and how many had NSAIDs, and which NSAIDs were they? How were either steroid injection or oral NSAIDs decided as the intervention of choice? Were there any differences in the results between those who had steroid injections compared to those who had NSAIDs?

Results and Table 4: Please can the authors clarify or comment on the clinical and sonographic correlation for synovitis which appear to be missing.

Limitations: Of course, as this has no control arm with using placebo, for example, the accuracy of the reported clinical outcomes may have a placebo effect, particularly if they had local treatment. Or the fact that if the dactylitis reduce in size, patients may see the physical size reduction and 'feel' less pain! Also, as mentioned above, the effect of a local intra-articular injection versus a systemic NSAIDs, and the perception of clinical response between the 2 therapies may be different. Please comment.

Reviewer 2 Report

An interesting observational study investigating the correlation between acute dactylitis and the sonographic lesions in psoriatic arthritis (PsA) patients. The study methods and results are well described as well as study limitations.

The study pointed to the correlation between clinical response of dactylitis to treatment and sonographic response characterized mainly by improvement in the extraarticular findings of flexor tenosynovitis and soft tissue edema.

My comments are

  1. In the introduction please update the reference regarding the European League Against Rheumatism (EULAR) recommendation to the most update one and not 2015.
  2. Please clarify how well controlled PsA was diagnosed “otherwise well-controlled PsA”experimemtal section . – was it according to the DAPSA score? It mainly addresses articular involvement. Data regarding enthesitis and axial involvement should be added as well as data regarding the duration of psoriasis, PASI score and nail involvement. The differentiation between DAPSA remission and low disease activity should be added as well.
  3. Regarding the fifteen patients who developed more than one episode of dactylitis, were the results adjusted for the concurrent treatment with steroids or NSAIDS prescribed at baseline for the first dactylitis event?
  4. Please provide the analysis of US findings adjusted to baseline treatment with cDMARDS, steroids as the treatment could have affected the US findings  
  5. Which variables were included in the multiple conditional logistic regression analysis? Did they included the additional data required in comment 2? Or treatment with cDMARDS? Steroids?